# A longitudinal assessment of the antibody response to SARS-CoV-2 infection in the New Mexican population

Frances M. Twohig[1]☯, Tonilynn M. Baranowski[1]☯, Chunyan Ye[1], Michelle Harkins[2], Steven B. Bradfute🄳[1]*

**1** Department of Internal Medicine, Center for Global Health, University of New Mexico Health Sciences Center, Albuquerque, New Mexico, United States of America, **2** Department of Internal Medicine, University of New Mexico Health Sciences Center, Albuquerque, New Mexico, United States of America

☯ These authors contributed equally to this work.
* sbradfute@salud.unm.edu

## Abstract

While many studies have assessed immune responses to SARS-CoV-2 infection, none have studied functional antibody responses before and after vaccination of exposed patients in New Mexico in the United States. Here, we evaluate antibody binding, antibody neutralization, and antibody dependent cell-mediated cytotoxicity (ADCC) responses from convalescent patients between September 2020 and April 2021. Our results indicate that binding, neutralizing, and ADCC titers remained durable over an estimated 4-month period or were boosted by vaccination. Antibody binding titer stability was comparable to that of antibodies against four common viruses. Hispanic and Latino responses were similar to non-Hispanic/Latino responses in this cohort. Overall, these data shed light on functional antibody responses to SARS-CoV-2 in pre-alpha variant waves in New Mexico.

## Introduction

Severe acute respiratory syndrome coronavirus 2 (SARS-CoV-2) is the etiological agent of coronavirus disease 2019 (COVID-19), which has infected over 775 million people globally, as of June 16th, 2024, and has caused over 7 million deaths [1]. In the United States (U.S.), non-Hispanic American Indian or Alaska Native, non-Hispanic Black, and Hispanic or Latino patients were disproportionately affected by the COVID-19 pandemic, when compared to non-Hispanic White patients [2]. Additionally, higher proportions of non-Hispanic Black and Hispanic people were hospitalized compared to the overall population [3]. Notably, New Mexico is one of only six majority-minority states in the U.S., meaning that non-Hispanic White people make up less than 50% of the population and, according to the 2020 U.S. Census, Hispanic or Latino people make up 47.7% of the population in New Mexico [4]. However, to the

**Data availability statement:** We have deposited our data set to Dryad: http://datadryad.org/share/1DxzK9y9c_IdS4QU9F2asirTrmQEQ-pzDwtLqSP863s.

**Funding:** UNM intramural grant CTSC003-11. The funders had no role in study design, data collection and analysis, decision to publish, or preparation of the manuscript. Materials were obtained through the 203298 Covid 19 Research Fund.

**Competing interests:** The authors have declared that no competing interests exist.

best of our knowledge, the polyfunctional antibody responses to SARS-CoV-2 in the New Mexican population has not been elucidated.

Most patients infected with SARS-CoV-2 produce Immunoglobulin G (IgG) and Immunoglobulin M (IgM) antibodies, primarily specific to the viral spike (S) and nucleocapsid (N) proteins, within 2 weeks after the onset of symptoms [5]. Previous studies have found that SARS-CoV-2 infection elicits an IgG antibody response to the S protein in patients that remains relatively stable for 5−6+ months, though the range of antibody titers varies greatly [6]. However, the stability of antibody responses against SARS-CoV-2 compared to other viruses is largely unknown.

Though it is crucial to measure antigen binding antibody titers in response to viral infection, antibody functions are key indicators of whether the antibodies will aid in protection against future infection and severe disease because they aid in viral clearance. It has been well documented that SARS-CoV-2 infection elicits neutralizing antibodies (NAbs), which primarily target the receptor binding domain (RBD) of the SARS-CoV-2 S protein to block the association of the RBD with angiotensin-converting enzyme 2 (ACE2) on host cells, preventing viral entry into host cells [7]. Khoury et al., 2021 [8] identifies NAbs against SARS-CoV-2, elicited from both vaccination and natural infection, as a predictor for immune protection against future symptomatic SARS-CoV-2 infection, with the neutralization level for 50% protection being 20.2% of the mean convalescent level from mild infection and 3% of the mean convalescent level from severe infection. These protective responses are suggested to wane as soon as 6-months post-infection [9,10] or over the course of 8 months post-infection [6,9]. Additionally, non-NAbs have been shown to be able to contribute to SARS-CoV-2 anti-viral activities through the elimination of infected cells by fragment crystallizable (Fc)-mediated effector functions, including antibody-dependent cell-mediated cytotoxicity (ADCC) [11,12]. ADCC is a cytotoxic mechanism by which the Fc region of IgG antibodies bound to antigens on the surface of an infected cell is recognized by the FcγR of an effector cell, mainly natural killer (NK) cells, resulting in release of cytotoxic granules by the NK cell that kill the infected cell [13]. Previous studies indicate that the ADCC response elicited by natural infection begins to wane at 3 months post-infection [14,15], though there are few SARS-CoV-2 longitudinal antibody studies which assess the ADCC response in SARS-CoV-2 experienced patients.

Our study provides a thorough analysis of SARS-CoV-2 antibody responses in an under-studied New Mexican population prior to the emergence of variants of concern, while comparing vaccinated and unvaccinated patients, further understanding the uncommonly studied ADCC response, and comparing antibodies elicited by other viruses to those elicited by SARS-CoV-2 in our patient cohort. Together, our data sheds light on antibody responses to SARS-CoV-2 infection during the early stages of the COVID-19 pandemic in the New Mexican population.

## Materials and methods

### Human subjects

The Institutional Review Board at the University of New Mexico approved all protocols and patient written informed consent (protocol 20-179, 28-04-2020 through

19-08-2024; minors were not enrolled). Serum, plasma, and peripheral blood mononuclear cells (PBMCs) were collected from PCR positive SARS-CoV-2 convalescent patients. Ethnicity, vaccine status, and co-morbidities were self-reported by each patient. Blood was drawn at least 28 days post-infection with a mean of 5.1 months +/- 2.47 months post-infection for draw 1, and at 4.4 months +/- 2 months post-draw 1 for draw 2. The self-reported vaccines received by patients after draw 1 were BNT162b2, mRNA-1273, or Janssen COVID-19. Blood was collected into BD Vacutainer® CPT™ Cell Preparation Tubes with Sodium Heparin (Cat. No. 362753) and BD Vacutainer® SST™ Blood Collection Tubes (Cat. No. 367986) and centrifuged within two hours of collection at 1500 xg for 30 minutes at room temperature. Samples were collected and immediately transferred into cryovials for storage at −80°C for use in antibody assays.

## Cell lines

Jurkat-Lucia™ NFAT-CD16 cells (InvivoGen, Cat. No. Jktl-nfat-cd16) were quickly thawed in 37°C water bath after receiving from InvivoGen (USA). The cells were transferred into 15 mL of pre-warmed test medium (Iscove's Modified Dulbecco's Medium (IMDM) containing 2mM L-glutamine and 25mM HEPES (Gibco, Cat. No. 12440046), supplemented with 10% heat-inactivated fetal bovine serum (FBS), 100 U/mL penicillin and 100 µg/mL streptomycin at 1% of medium (P/S) (Cat. No. 15140-122)) with the addition of Normocin™ and centrifuged at 800 rpm for 5 minutes. The supernatant was discarded, and the cells were resuspended in 1 mL of test medium and Normocin™. Cells were passaged every 3–5 days and discarded after passage 20. The medium was supplemented with 10 µg/mL of Blastocidin (InvivoGen, Cat. No. anti-bl-05) and 100 µg/mL of Zeocin™ (InvivoGen, Cat. No. anti-zn-05) every other passage after resting for two passages in the test medium with Normocin™.

293-SARS2-S cells, which are HEK293-derived cells overexpressing the Wuhan-Hu-1 SARS-CoV-2 S gene, (InvivoGen, Cat. No. 293-cov2-s) were cultured in growth medium (Dulbecco's Modified Eagle's Medium (DMEM) (Cat. No. 11965-084) from Gibco (USA) containing 2mM L-glutamine and 25mM HEPES, and supplemented with 10% FBS, 1% P/S, and 100 µg/mL Normocin™) with the addition of 10 µg/mL of Blastocidin. Expression of the Wuhan-Hu-1 SARS-CoV-2 S protein was validated by the manufacturer through flow cytometry. Upon receiving the cells from InvivoGen (USA), they were quickly thawed in a 37°C water bath and transferred into 15 mL of pre-warmed growth medium and centrifuged at 1200 rpm for 5 minutes. The supernatant was discarded, and the cells were resuspended in 1 mL of growth medium. 293-SARS-CoV-2-Spike cells were passaged every 2–4 days, or when 80% confluency was reached, and discarded after passage 20.

## Viruses and antigens

The Wuhan strain of SARS-CoV-2 was the most common circulating strain during the time of infection of our patients and the dominant circulating strain did not change during our study timeline. As such, viruses and antigens homologous with the Wuhan strain were utilized throughout this study.

## SARS-CoV-2 ELISA

Serum IgG titers against the SARS-CoV-2 Spike protein were measured using Recombivirus Human anti-SARS-CoV-2 (COVID-19) Spike protein 1 (S1) IgG (Alpha Diagnostic International, Cat. No. RV-405200). The ELISA was performed according to the manufacturer's instructions and with provided standards. Serum was initially run at a dilution of 1:2000 and samples were run in duplicate. If the optical density (O.D.) of the diluted sample did not fit within the standard curve, the sample dilutions were adjusted accordingly to fit within the standard values. O.D. values were input into the standard curve to calculate the concentration of antibody (U/mL). This value was multiplied by the sample dilution factor to calculate the final concentration.

## HCoV-229E ELISA

Serum IgG titers against the HCoV-229E Spike protein were measured using Recombivirus Human anti-HCoV-229E S1 IgG ELISA Kits (Cat. No. RV-406100) obtained from Alpha Diagnostic International (USA). The ELISAs were performed

according to the manufacturer's protocol, utilizing a serum dilution of 1:1000. If the sample's O.D. was outside of the range of the standards provided with the kit, the protocol was performed again with higher or lower dilutions, respectively. Final antibody concentrations were calculated as described previously.

### Measles ELISA

Serum IgG titers against Measles Virus were measured using the EUROIMMUN Anti-Measles Virus ELISA (IgG) test kit (Cat. No. EI 2610-9601 G). The ELISA was performed according to the manufacturer's protocol with provided standards and controls. Serum samples were run in duplicate at a dilution of 1:101. Final antibody concentrations were calculated as described previously.

### Cytomegalovirus ELISA

Serum IgG titers against cytomegalovirus were measured using the EUROIMMUN Anti-Cytomegalovirus ELISA (IgG) test kit (Cat. No. EI 2570-9601 G). The ELISA was performed according to the manufacturer's protocol and with provided standards. Serum was run at a dilution of 1:101 and samples were run in duplicate. Final antibody concentrations were calculated as described previously.

### Adenovirus ELISA

Serum IgG titers against Adenovirus were measured using the EUROIMMUN Anti-Adenovirus (strain adenoid 6) ELISA (IgG) test kit (Cat. No. EI 2680-9601 G). The ELISA was performed according to the manufacture's protocol and with provided standards. Serum was run at a dilution of 1:101 and samples were run in duplicates. Final antibody concentrations were calculated as described previously.

### Plaque-reduction neutralization test (PRNT)

Vero E6 cells were seeded onto a 12-well plate and incubated at 37°C for 12–24 hours until cells reached 90% confluency. Stock virus was prepared by diluting 50–100 PFU SARS-CoV-2 in 200 μL virus growth medium (VGM), containing Minimum Essential Medium (MEM) with 2.5% heat inactivated FBS. Serial dilutions of serum in VGM were added to diluted virus at equal volumes. This mixture was incubated at 37°C for 1–1.5 hours. After incubation, 400 μL of the serum-virus mixture and virus controls were added to the Vero E6 cells and were incubated at 37°C. After 2 hours, the media was aspirated, and the cells were washed with PBS. Viral overlay medium (VOM) containing equal volumes of 2% agarose and 2X Modified Eagle Medium concentrate, and supplemented with 5% FBS and 2X antibiotics, was added to the cells and incubated at 37°C for 2–3 days. After 2–3 days, the VOM was aspirated and the plate was fixed with 4% formaldehyde overnight at 4°C. The fixative reagent was aspirated, and the cells were stained with 0.5% crystal violet for 1–2 minutes. The plate was washed under running water. After the plate dried, the plaques were quantified for each serum dilution. The serum dilution that elicited an 80% reduction in plaques when compared to the virus control plate indicated the PRNT80 value.

### S-specific antibody-dependent cellular cytotoxicity (ADCC) assay

Plasma was diluted at 1:16 in test media and plated into a 96-well flat-bottom tissue culture treated plate in triplicates. Effector cells, Jurkat-Lucia™ NFAT-CD16, were diluted in pre-warmed test media and added to the co-incubated plasma and target cells at $2 \times 10^5$ cells per well. After incubating the samples for 18 hours at 37°C and 5% $CO_2$, the bioluminescence detection reagent, QUANTI-Luc™ (InvivoGen, Cat. Code. Rep-qlc1), and the samples were equilibrated to room temperature. 50 μL of the cell supernatant was transferred into a 96-well white opaque plate, and 50 μL QUANTI-Luc™ was added to each well. After gently mixing the samples by tapping the plate, bioluminescence was measured immediately on the BioTek Synergy HTXNeo2 Multi-Mode Microplate Plate Reader Analyzer. Controls included target cells only,

target cells and QUANTI-Luc™ only, no effector cells, no plasma, test media and QUANTI-Luc™ only, and SARS-CoV-2 negative plasma. The fold of induction (FOI) was calculated by using relative light units in the following equation:

$$FOI = \frac{induced - background}{SARS-CoV-2\ negative\ plasma\ control - backround}$$

## Statistical analysis

All statistical analyses were performed using GraphPad Prism version 10.2.3. Kruskal-Wallis rank sum tests were performed to assess significance between groups and Dunn's tests were performed alongside to adjust for multiple comparisons. Wilcoxon matched-pairs signed rank tests were used for analysis within each group. Student T-tests were performed for Table 2. Additional statistical methods were included in the figure legends.

## Results

Between September 23rd, 2020 and April 16th 2021, we recruited a diverse cohort of 45 patients, with 62.22% of patients being Hispanic or Latino, 31.11% being White Non-Hispanic, 4.44% being American Indian/Alaskan Native Non-Hispanic, and 2.22% being Asian/Pacific Islander Non-Hispanic, who were infected with SARS-CoV-2 at least 28 days prior to admission to the study (Table 1). The patients in our cohort tested positive for SARS-CoV-2 infection by PCR between March 2nd, 2020, and January 28th, 2021, and experienced a range of disease from asymptomatic to severe (with hospitalization). We conducted an observational immune response study with a two blood draws: one at enrollment (5.1 months +/- 2.47 months post infection, hereafter referred to as "draw 1") and a second blood draw at 4.4 months +/- 2 months after draw 1 (hereafter referred to as "draw 2").

**Table 1. Demographics and clinical characteristics of our patient cohort.**

| Patient Demographics and Clinical Characteristics | Number of patients (n) | Percentage of patients (%) |
|---|---|---|
| **Sex** | | |
| Male | 20 | 44% |
| Female | 25 | 56% |
| **Age** | | |
| <40 | 21 | 47% |
| 41–60 | 17 | 38% |
| >60 | 7 | 16% |
| **Ethnicity** | | |
| Hispanic or Latino | 28 | 62.22% |
| White, Non-Hispanic | 14 | 31.11% |
| American Indian/Alaskan Native, Non-Hispanic | 2 | 4.44% |
| Asian/Pacific Islander, Non-Hispanic | 1 | 2.22% |
| **Hospitalization** | | |
| Out-patient | 31 | 69% |
| In-patient | 14 | 31% |
| **Vaccination status (3–6 months post-enrollment)** | | |
| Vaccinated | 15 | 33% |
| Unvaccinated | 22 | 49% |
| Unknown | 8 | 18% |

Vaccines against SARS-CoV-2 were made available between draw 1 and draw 2 of our study. Thus, we split our patients into two groups: those who were vaccinated between draw 1 and draw 2 (henceforth referred to as "vaccinated") and those who were not vaccinated between draw 1 and draw 2 (henceforth referred to as "unvaccinated"). 11 unvaccinated and 14 vaccinated patients returned for draw 2 and were included in our analysis. Most of the vaccinated patients were vaccinated with either BNT162b2 (Pfizer-BioNTech) or mRNA-1273 (Moderna), with only two patients being vaccinated with Janssen COVID-19. Individuals in this cohort also reported a range of co-morbidities. These included diabetes mellitus, hypertension, chronic lung disease, various cancer types, asthma, and pregnancy. Due to the sample size, sub-analyses by vaccine type or co-morbidity is not possible for this study.

The S1 protein, containing the immunodominant RBD of SARS-CoV-2 [16] was used to identify potential changes in IgG antibody titers in our cohort over time. As shown in Fig 1, at draw 1, serum S1-specific IgG titers varied significantly with titers ranging from 385 U/mL to 47,099 U/mL. At draw 2, IgG titers were comparable to draw 1 in unvaccinated patients, ranging from 553 U/mL to 45,308 U/mL. Conversely, there was a significant increase in IgG titers in vaccinated patients between draw 1 and draw 2, with draw 2 titers ranging from 1,935 U/mL to 1,594,286 U/mL. When comparing draw 2 titers between unvaccinated and vaccinated, we found that there was a significant difference in IgG concentration, with vaccinated patients having on average higher S1-specific IgG titers; there was no significant difference between unvaccinated and vaccinated IgG titers at draw 1. These data suggest that the magnitude of the antibody responses to S1 is dependent on the patient and time post-infection, which resulted in a range of IgG titers elicited through natural

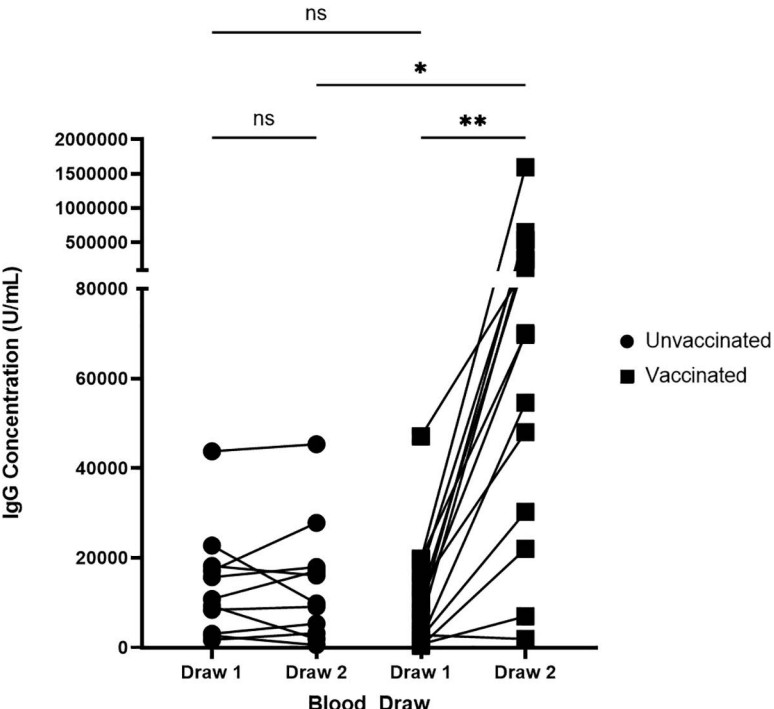

**Fig 1. ELISA IgG antibody concentrations targeting SARS-CoV-2.** Serum was collected 5.1 months +/- 2.47 months post infection for draw 1 and 4.4 months +/- 2 months after draw 1 for draw 2 and was analyzed for SARS-CoV-2 S1 IgG binding utilizing an ELISA. IgG concentrations were calculated utilizing the trendline equation created using the standards included in the kit. Left, individuals who were not vaccinated between the first and second draw (N = 11). Right, individuals vaccinated between draw 1 and draw 2 (N = 14). ***P < 0.001; **P < 0.01; *P < 0.05 using a Wilcoxon matched-pairs signed rank test for analysis within each group and Kruskal-Wallis tests for analysis between unvaccinated and vaccinated groups.

SARS-CoV-2 infection in our cohort. Additionally, vaccination appears to boost S1-specific IgG antibody titers while there was no significant decline in IgG concentration between draw 1 and draw 2 in the unvaccinated group. This indicates that S1-specific antibody titers, on average, remained stable for at least 4.4 months in our patient cohort.

To further evaluate the stability of S1-specific IgG antibodies in our SARS-CoV-2 cohort, we measured IgG antibody titers of common viruses, including AdV, CMV, measles virus, and HCoV-229E (Fig 2). These viruses either cause persistent infection (AdV and CMV) [17,18], produce long-lived antibody responses post-vaccination (measles virus) [19] or are reoccurring in the population (HCoV-229E) [20], thus there is a high likelihood of our patients having circulating antibodies against these viruses. Unvaccinated patient serum was tested for IgG antibodies specific for each virus, at both draw 1 and draw 2. When comparing the fold change across all viruses tested for, there was no significant difference, indicating that the stability of IgG titers elicited from natural SARS-CoV-2 infection was similar to that of other viruses in the same patient cohort over the same estimated 4.4-month period.

To assess functional antibody responses, we measured our patient's SARS-CoV-2-specific NAb titers through a plaque-reduction neutralization test (PRNT) using live virus (Fig 3). Both unvaccinated and vaccinated groups exhibited a range of NAb concentrations at draw 1, with 42% of patients having no detectable NAbs and the highest PRNT80 concentration reaching 1,280. Between draw 1 and draw 2, there was a significant increase in NAb titers in the vaccinated group, while there was no significant difference in the unvaccinated group. However, three unvaccinated patients did demonstrate an increase in NAb titers between draw 1 and draw 2, possibly indicating a second infection by SARS-CoV-2 between draws. Interestingly, the average NAb concentrations in the unvaccinated group remained stable, but the NAb concentrations in the vaccinated group at draw 2 were significantly greater than the unvaccinated group at draw 2. These data suggest that vaccination boosted NAb titers in our cohort and support the effectiveness of vaccination to produce SARS-CoV-2-specific NAbs.

Fc-mediated functional antibody responses against SARS-CoV-2 were also measured using an S-specific ADCC assay for both unvaccinated and vaccinated patients (Fig 4). We detected ADCC activity (FOI > 1) against S in all patients except for three at draw 1, with many of the patients having low ADCC activity. In our unvaccinated group we did not observe significant differences in the FOI values between draw 1 and draw 2. However, in our vaccinated group, there was a

## Fold Change in IgG concentration

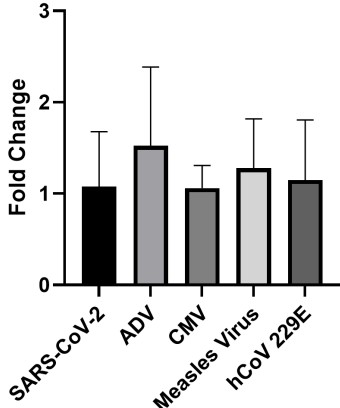

**Fig 2. Comparison of SARS-CoV-2 antibody concentration stability to other common viruses.** Serum was collected 5.1 months +/- 2.47 months post infection for draw 1 and 4.4 months +/- 2 months after draw 1 for draw 2. IgG binding to AdV, CMV, measles virus, HCoV-229E Spike protein, and SARS-CoV-2 S1 was assessed through a series of ELISAs. Only patients who were not vaccinated (N = 11) against SARS-CoV-2 at draw 2 were used in this analysis. IgG concentrations were calculated utilizing the trendline equation created using the standards included in each kit. Fold change in IgG concentration between draw 1 and draw 2 was calculated for each patient. ***P < 0.001; **P < 0.01; *P < 0.05 using Kruskal-Wallis tests.

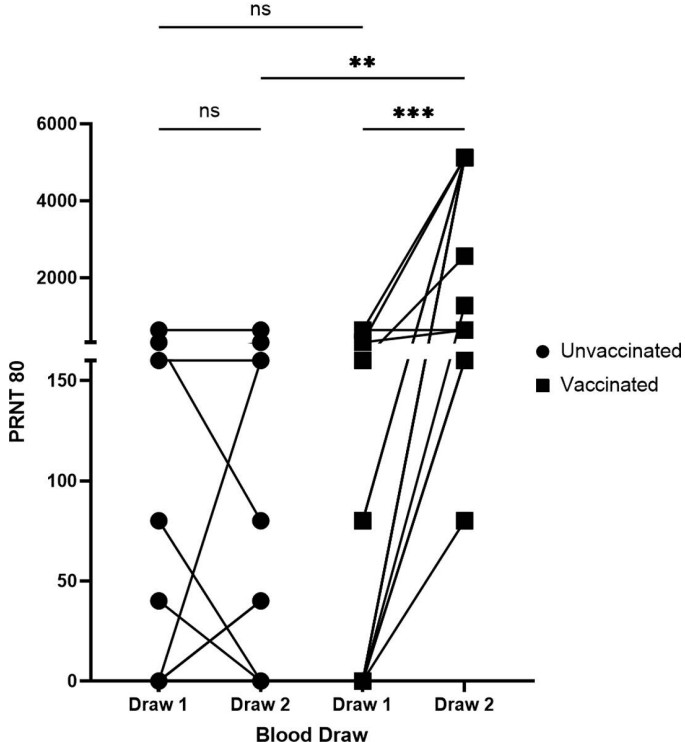

**Fig 3. Neutralizing antibody titers to SARS-CoV-2 over time.** Serum was collected 5.1 months +/- 2.47 months post infection for draw 1 and 4.4 months +/- 2 months after draw 1 for draw 2. Neutralization of SARS-CoV-2 live virus was assessed through a PRNT. Left, individuals who were not vaccinated between the first and second draw (N = 11). Right, individuals vaccinated between draw 1 and draw 2 (N = 14). ***P < 0.001; **P < 0.01; *P < 0.05 using a Wilcoxon matched-pairs signed rank test for analysis within each group and Kruskal-Wallis tests for analysis between unvaccinated and vaccinated groups.

significant increase in S-specific ADCC activity between draw 1 and draw 2, suggesting that vaccination boosts ADCC responses in our cohort.

With the knowledge that Hispanic or Latino patients are disproportionately affected by SARS-CoV-2, we compared antibody responses of Hispanic or Latino and non-Hispanic patients to SARS-CoV-2 in both patients who were unvaccinated and vaccinated between draw 1 and draw 2, as depicted in Table 2. There were no significant differences in the antibody titers or antibody functionality between Hispanic and non-Hispanic patients in our cohort.

## Discussion

Our work aimed to elucidate the antibody response to SARS-CoV-2 in the New Mexican population through an observational immune study. Throughout the COVID-19 pandemic, Non-Hispanic American Indian/Alaska Native, Non-Hispanic Black or African American, and Hispanic or Latino patients have suffered disproportionate rates of infection by SARS-CoV-2 as well as hospitalization and death due to SARS-CoV-2, when compared to Non-Hispanic White patients [2,3]. New Mexico is a majority-minority state, with 47.7% of the population being Hispanic or Latino [4], making it important to understand the immune responses in the New Mexican population. To do this, we recruited a diverse cohort of 45 patients who had previously been infected with SARS-CoV-2 and evaluated antibody responses at two different time points: 5.1 months +/- 2.47 months post infection (draw 1), and 4.4 months +/- 2 months after draw 1 (draw 2). At the time of the first

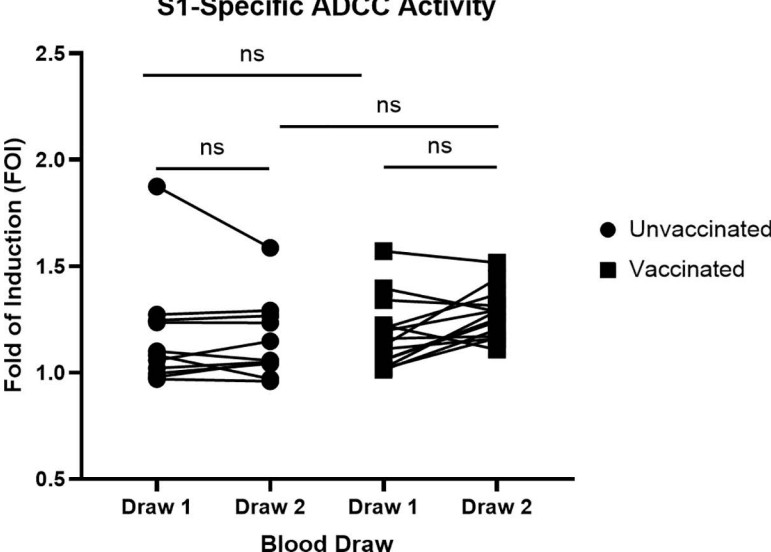

**Fig 4. Measuring S-specific ADCC functions of antibodies.** Plasma was collected 5.1 months +/- 2.47 months post infection for draw 1 and 4.4 months +/- 2 months after draw 1 for draw 2. ADCC activity was assessed using an ADCC luciferase reporter assay. FOI was calculated using relative light units and the following equation: $FOI = \frac{induced-background}{SARS-CoV-2\ negative\ plasma\ control-backround}$. Left, individuals who were not vaccinated between the first and second draw (N=11). Right, individuals vaccinated between draw 1 and draw 2 (N=14). ***P<0.001; **P<0.01; *P<0.05 using a Wilcoxon matched-pairs signed rank test for analysis within each group and Kruskal-Wallis tests for analysis between unvaccinated and vaccinated groups.

**Table 2. Comparing results of hispanic patients to non-Hispanic patients.**

| Experiment | Hispanic | Non-Hispanic | P-Value |
|---|---|---|---|
| **IgG Concentration Draw 1 (U/mL)** | 13976.90 | 12058.70 | >0.9999 |
| **IgG Concentration Draw 2 (U/mL)** | 188885.26 | 94997.21 | 0.1261 |
| **Neutralization Draw 1 (PRNT 80)** | 194.07 | 140.00 | >0.9999 |
| **Neutralization Draw 2 (PRNT 80)** | 1606.32 | 1567.50 | >0.9999 |
| **ADCC Draw 1 (FOI)** | 1.21 | 1.19 | >0.9999 |
| **ADCC Draw 2 (FOI)** | 1.29 | 1.21 | >0.9999 |

IgG titers, Neutralizing antibody titers, and ADCC activity at draw 1 and draw 2 from Figs 1–4 were averaged for both Hispanic (N=27) and Non-Hispanic (N=18) patients our patient cohort, regardless of whether they returned for draw 2. A student T-Test was performed for each of these data sets.

draw, vaccines against SARS-CoV-2 were not yet available to the public, however they were by draw 2. As a result, the cohort was analyzed based on vaccine status.

SARS-CoV-2 S1-specific IgG titers ranged from 385 to 47,099 U/mL at draw 1 across patients, and from 553 U/mL to 45,308 U/mL at draw 2 in the unvaccinated group. As expected, vaccine-boosted patients had, on average, significantly higher IgG titers for draw 2 than the unvaccinated group. IgG titers in the unvaccinated group did not significantly change from draw 1 to draw 2, however, IgG titers significantly increased in the vaccinated group from draw 1 to draw 2. Several previous studies corroborate these findings and have demonstrated that anti-SARS-CoV-2 S1 IgG concentrations remained elevated for at least 3 months post-infection with modest declines between 3−8 months post-infection in the absence of reinfection or vaccination [6,9,21–23]. To better understand the short-term stability of SARS-CoV-2-specific IgG titers in our cohort, we also measured the IgG titers from draw 1 to draw 2 for viral antibodies commonly found in the population: AdV, CMV, HCoV-229E, and measles virus. We calculated the fold-change of IgG titers between

the unvaccinated group's blood draws 1 and 2 for each virus and compared the variation of these titers over our time course. Interestingly we did not see a significant difference in the short-term stability of antibodies elicited by natural SARS-CoV-2 infection versus other common viruses.

NAb responses exhibited a similar pattern to antibody binding between draw 1 and draw 2 of both the unvaccinated and vaccinated groups. There was no significant difference between draw 1 and draw 2 NAb titers in the unvaccinated group, while there was a significant increase in NAbs between draw 1 and draw 2 of the vaccinated group. Despite the average NAb titers remaining stable in the unvaccinated group, a few patients did have increased titers at draw 2. One possible reason for this may be reinfection. No patients reported symptomatic reinfection, however, the time course of this observational study was brief and protective antibody responses could have resulted in asymptomatic reinfection. We also saw that the unvaccinated group at draw 2 had significantly lower NAb titers than the vaccinated group. This is in agreement with a study demonstrating that vaccination against SARS-CoV-2 elicits an increase in NAbs, even after natural infection [24]. It is notable that 42% of our cohort lacked NAb responses against SARS-CoV-2 at draw 1, while a previous study indicated that >90% of seroconverters produced NAbs [21]. The difference may be attributable to the time post-infection, where our cohort was screened at on average about 5 months post-infection, and the previous study's cohort was screened during the acute phase of disease. Other longitudinal SARS-CoV-2 antibody studies have demonstrated that NAb titers after primary infection begin to wane after 6 weeks post-infection [9,10], with Anand et al., 2021 showing that only ~20% of convalescent patients had detectable NAbs at 8 months post-infection [9] but Dan et al., 2021 showing that 90% of patients had detectable neutralizing antibodies at 6–8 months post infection [6]. A decline in NAb titers over a 3-month period has also been described [25]. Our data largely agrees with previous studies as there was not a significant decline in antibody titers in our unvaccinated group between draw 1 and draw 2.

The SARS-CoV-2 S-specific ADCC responses largely followed the same trend as antibody binding and NAb titers. ADCC activity remained low, but detectable in all patients except for three and there was no significant change in ADCC responses between draw 1 and draw 2 in the unvaccinated group. Zedan et al., 2024, demonstrated that S-specific ADCC activity was induced within 1 month of infection or vaccination and remained detectable for ≥ 3 months [26]. Similarly, Lee et al., 2021, notes that S-specific ADCC activity remains detectable for up to 4 months post-infection in 94% of convalescent patients [27], while Anand et al., 2021, saw modest declines in ADCC activity between 6- and 31-weeks post-infection [9]. Our results largely corroborate these studies, as our patients did not experience a significant decline over an estimated 4.4-month period. The vaccinated group saw a significant increase in S1-specific ADCC activity at draw 2 when compared to draw 1, though there was not a significant difference between draw 2 of the vaccinated and unvaccinated groups. Several studies indicate that vaccination against SARS-CoV-2 triggers ADCC activity [26,28,29], in alignment with our results. Notably, we utilized a reporter cell line (Jurkat-Lucia™ NFAT-CD16) in our ADCC assay rather than primary NK cells. This allowed us to limit batch variability associated with primary NK cells. ADCC assays utilizing both the reporter cell line and primary NK cells rely on crosslinking of CD16 on the cell surface. In the case of our reporter cell line, this leads to a cytoplasmic signaling cascade resulting in the release of Lucia luciferase into the supernatant. The endpoint of ADCC assays using primary NK cells is cell death due to the release of cytotoxic granules. While endpoints of these assays differ, the widely used Jurkat-Lucia™ NFAT-CD16 cell line is less resource intensive and less variable than primary NK cells in ADCC assays, allowing for clearer comparisons across experiments.

Our study illustrates that patients in the New Mexican population who are vaccinated against SARS-CoV-2 after infection see a significant increase in IgG and NAb titers and ADCC activity against SARS-CoV-2. Conversely, in unvaccinated patients, antibody titers do not significantly change over this study. Additionally, our data indicates that the short-term stability of IgG titers to SARS-CoV-2 infection is comparable to the stability of IgG titers elicited by AdV, CMV, and HCoV-229E infection and measles virus vaccination. When comparing the results from each of the antibody studies performed in this study between Hispanic and Non-Hispanic patients, there were no significant differences. It is possible that a more comprehensive analysis with a larger cohort could reveal details regarding differences in antibody responses between

demographic groups, across longitudinal samples, or between vaccinated and unvaccinated groups across this study. The limited sample size in this study hindered our ability to perform further sub-group analyses, such as stratification by vaccine type or co-morbidity. Future studies with expanded cohorts would allow for relevant sub-group analyses.

This observational study largely agrees with current SARS-CoV-2 humoral immune studies. The New Mexican population is uniquely diverse, including groups that are historically underrepresented in biomedical research. By investigating the antibody response to SARS-CoV-2 in this region, we can gain valuable insights into potential disparities in antibody responses and vaccine efficacy, informing public health guidelines for understudied groups. This study may also function as an internal benchmark of the antibody response in the New Mexican population prior to the emergence of variants of concern. Overall, this study provides a thorough analysis of antibody concentration and functionality against SARS-CoV-2 in the New Mexican population.

## Author contributions

**Conceptualization:** Frances M. Twohig, Tonilynn M Baranowski, Steven S. Bradfute.

**Data curation:** Frances M. Twohig, Tonilynn M Baranowski, Chunyan Ye.

**Formal analysis:** Frances M. Twohig, Steven S. Bradfute.

**Funding acquisition:** Michelle Harkins, Steven S. Bradfute.

**Investigation:** Frances M. Twohig, Tonilynn M Baranowski, Chunyan Ye, Michelle Harkins.

**Methodology:** Frances M. Twohig, Tonilynn M Baranowski, Chunyan Ye.

**Project administration:** Steven S. Bradfute.

**Resources:** Michelle Harkins, Steven S. Bradfute.

**Supervision:** Steven S. Bradfute.

**Validation:** Frances M. Twohig.

**Writing – original draft:** Frances M. Twohig.

**Writing – review & editing:** Frances M. Twohig, Tonilynn M Baranowski, Chunyan Ye, Michelle Harkins, Steven S. Bradfute.

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
