## [Editor Report · Decision Letter 0]

PONE-D-24-35337

A Longitudinal Assessment of the Antibody Response to SARS-CoV-2 Infection in the New Mexican Population

PLOS ONE

Dear Dr. Bradfute,

Thank you for submitting your manuscript to PLOS ONE. After careful consideration, we have decided that your manuscript does not meet our criteria for publication and must therefore be rejected.

I am sorry that we cannot be more positive on this occasion, but hope that you appreciate the reasons for this decision.

Kind regards,

Batool Mutar Mahdi

Academic Editor

PLOS ONE

Additional Editor Comments:

The article was well written but it is preferable to publish in your regional country

- - - - -

---

## [Author Response · Author response to Decision Letter 1]

18 Dec 2024

We did not receive any feedback from any reviewers, suggesting that none were obtained during the 3.5 month interval that PLOS ONE was in receipt of our manuscript. The only comments we received from the academic editor was:

“After careful consideration, we have decided that your manuscript does not meet our criteria for publication and must therefore be rejected”

and

“The article was well written but it is preferable to publish in your regional country.”

Our responses:

1) There are no criteria in the PLOS ONE “Criteria for Publication” that includes reasoning that some studies are better served being published in a “regional country.” This reasoning appears to be a clear violation of PLOS ONE policy.

2) Our study occurred in New Mexico, which has been a state in the United States of America since 1912. (Our article clearly states that New Mexico is part of the US in the introduction; we have added this as well to the absract). It was the 47th state, and was admitted before Arizona, Alaska, and Hawaii. The idea that we should be publishing in our “regional country” would mean we should be publishing in PLOS ONE, which is based in the US. However, in case there is a requirement for proving that a study in New Mexico deserves to be published in the US, here are some fun facts about New Mexico:

a) New Mexico is the 5th largest state by size in the US.

b) New Mexico, despite being 37th in population, has the most human hantavirus cases in the US.

c) New Mexico is home to 2 US national labs (Sandia National Laboratories and Los Alamos National Laboratory).

d) New Mexico is home to 3 US air force bases (Cannon, Holloman, and Kirtland). Many of the original stealth fighter aircraft were housed at Holloman.

e) New Mexico is the 2nd largest oil-producing state in the US and is 7th in the US in wind power.

f) The first nuclear bomb was developed and tested in New Mexico.

g) New Mexico is a major filming site in the US, known for such shows as “Breaking Bad,” “Better Call Saul,” and “Oppenheimer.”

h) The capital of New Mexico, Santa Fe, is the oldest state capital in the US, founded in 1610.

i) New Mexico is one of 7 states that have a majority-minority population.

j) New Mexico has the 3rd-highest percentage of Native American citizens of any state in the US.

k) New Mexico produces 33% of chile and 29% of pecans in the US.

l) New Mexico hosts an annual hot air balloon fiesta that is the largest in the world.

Therefore, we feel that our manuscript should be reviewed and is suitable for publication in PLOS One.

---

## [Decision Letter · Decision Letter 1]

PONE-D-24-35337R1A Longitudinal Assessment of the Antibody Response to SARS-CoV-2 Infection in the New Mexican PopulationPLOS ONE

Dear Dr. Bradfute,

Thank you for submitting your manuscript to PLOS ONE. After careful consideration, we feel that it has merit but does not fully meet PLOS ONE’s publication criteria as it currently stands. Therefore, we invite you to submit a revised version of the manuscript that addresses the points raised during the review process.

I have read the manuscript particularly carefully and I very much agree with reviewer 1 regarding the structural flaws in the manuscript; I encourage the authors to provide a comprehensive and evidence-based response and to incorporate the relevant changes and additions.

We look forward to receiving your revised manuscript.

Kind regards,

José Ramos-Castañeda, M.Sc., Ph.D

Academic Editor

PLOS ONE

“UNM intramural grant CTSC003-11”

3. Please note that funding information should not appear in the Acknowledgments section or other areas of your manuscript. We will only publish funding information present in the Funding Statement section of the online submission form. Please remove any funding-related text from the manuscript.

4. We note that your Data Availability Statement is currently as follows:

“All relevant data are within the manuscript and its Supporting Information files”

Additional Editor Comments (if provided):

Reviewers' comments:

Reviewer's Responses to Questions

**Comments to the Author**

1. If the authors have adequately addressed your comments raised in a previous round of review and you feel that this manuscript is now acceptable for publication, you may indicate that here to bypass the “Comments to the Author” section, enter your conflict of interest statement in the “Confidential to Editor” section, and submit your "Accept" recommendation.

Reviewer #1: (No Response)

Reviewer #2: All comments have been addressed

2. Is the manuscript technically sound, and do the data support the conclusions?

Reviewer #1: Partly

Reviewer #2: Yes

3. Has the statistical analysis been performed appropriately and rigorously? 

Reviewer #1: No

Reviewer #2: Yes

4. Have the authors made all data underlying the findings in their manuscript fully available?

Reviewer #1: Yes

Reviewer #2: Yes

5. Is the manuscript presented in an intelligible fashion and written in standard English?

Reviewer #1: Yes

Reviewer #2: Yes

6. Review Comments to the Author

Reviewer #1: This manuscript addresses the antibody response (binding, neutralizing, and ADCC) to SARS-CoV-2 infection and vaccination in a New Mexican cohort, with attention to ethnic representation. The study answers relevant questions about the duration of humoral immunity response and Fc-mediated functions post-infection and post-vaccination and adds comparisons with other viruses.

While the manuscript is based on sound scientific methods and presents results that are broadly consistent with current literature, there are several critical limitations and structural issues that must be addressed before the manuscript can be considered for publication.

Limitations by Section

1. Introduction

The introduction section inappropriately includes extensive information regarding the study methodology and main results. Specifically:

The recruitment timeline, patient demographics, and cohort description (lines 36-53).

Key results related to IgG, NAb, and ADCC findings (lines 91-110) are presented here.

Information about the circulating SARS-CoV-2 strain (Wuhan) and the study design (two blood draws) should be moved to the Materials and Methods section.

The introduction should be refocused on providing background, identifying the knowledge gap, and stating the study objective without introducing data or methods.

2. Materials and Methods

Incomplete reporting of key experimental details:

The authors mention using HEK293 cells overexpressing the Wuhan-Hu-1 S gene for ADCC assays but do not report the validation of the tecnic for the expression of the S protein (e.g., by flow cytometry or Western blot). Given the reliance on these target cells, internal validation should be either performed or referenced.

Lack of information on statistical correction:

There is no mention of adjustment for multiple comparisons (e.g., Bonferroni or FDR) despite multiple statistical tests (IgG, NAb, ADCC across groups and timepoints).

3. Results

While technically sound, the results section would benefit from a subgroup analysis based on vaccine type (Pfizer-BioNTech, Moderna, Janssen), as this could influence immune responses. This omission limits the interpretability of vaccination-related findings.

There is no mention of comorbidities in the population of the study.

The sample size in some subgroups (e.g., unvaccinated N=11, vaccinated N=14) is modest, which restricts the statistical power for detecting differences, particularly in ADCC data where variability is expected.

In the results, you grouped participants simply as "vaccinated" without distinguishing whether there were differences among those who received different types of vaccines.

The manuscript would be stronger if you mentioned that you attempted to compare vaccine types, or if you acknowledged that the sample size limited the ability to perform sub-analyses by vaccine type. This is important because each vaccine elicits distinct humoral and Fc-mediated immune responses.

4. Discussion

The discussion does not fully address some key limitations:

There is no discussion on the lack of vaccine-type stratification in the vaccinated group.

The authors acknowledge the limited cohort size in the ethnic comparison but should also emphasize how this impacts statistical power across all comparisons (e.g., antibody functional assays).

The discussion would also benefit from a clearer statement regarding the scope and applicability of the findings. For example, the authors should elaborate on how their results could be generalized (or not) to other populations or settings beyond the New Mexican cohort, and the potential relevance of these findings to current SARS-CoV-2 variants or vaccination strategies.

The discussion could also benefit from reflecting on the potential implications of using reporter cell lines (Jurkat-Lucia™ NFAT-CD16) versus primary NK cells in ADCC assays, which is relevant for interpreting effector function results.

Conclusion

This manuscript provides valuable data on immune responses to SARS-CoV-2 in a minority-majority U.S. population. However, several structural and methodological issues need to be resolved to meet the journal's standards.

Reviewer #2: The authors developed a project with the aim of evaluating the immune response, measured in various ways, of people with COVID-19 and later in the latter period, comparing those who had received the vaccine and those who had not.

The work is interesting even in the current state of COVID-19, given the various trials to evaluate the immune response, and the opportunity to observe the behavior in the first phase of vaccination.

The laboratory techniques are very well described. The statistical analysis and results are adequately presented.

Perhaps it would be helpful if the authors, if they have the information, could comment on whether their cohort used only one type of vaccine, or several, and if so, which ones were available to them at that time.

Table 2 includes an unusual column interpreting p values as non-significant. Perhaps the column with the p value could be left alone and left to each reader to interpret it.

Some journals require that the p value have only a few values after the decimal point; it is worth review it.

7. PLOS authors have the option to publish the peer review history of their article (what does this mean? ). If published, this will include your full peer review and any attached files.

**Do you want your identity to be public for this peer review?** For information about this choice, including consent withdrawal, please see our Privacy Policy .

Reviewer #1: No

Reviewer #2: No

---

## [Author Response · Author response to Decision Letter 2]

3 Jun 2025

A response to reviewers section is included in this submission.

---

## [Decision Letter · Decision Letter 2]

A Longitudinal Assessment of the Antibody Response to SARS-CoV-2 Infection in the New Mexican Population

PONE-D-24-35337R2

Dear Dr. Bradfute,

We’re pleased to inform you that your manuscript has been judged scientifically suitable for publication and will be formally accepted for publication once it meets all outstanding technical requirements.

Kind regards,

José Ramos-Castañeda, M.Sc., Ph.D

Academic Editor

PLOS ONE

Additional Editor Comments (optional):

Reviewers' comments:

Reviewer's Responses to Questions

**Comments to the Author**

1. If the authors have adequately addressed your comments raised in a previous round of review and you feel that this manuscript is now acceptable for publication, you may indicate that here to bypass the “Comments to the Author” section, enter your conflict of interest statement in the “Confidential to Editor” section, and submit your "Accept" recommendation.

Reviewer #1: All comments have been addressed

Reviewer #2: All comments have been addressed

2. Is the manuscript technically sound, and do the data support the conclusions?

Reviewer #1: Yes

Reviewer #2: (No Response)

3. Has the statistical analysis been performed appropriately and rigorously? 

Reviewer #1: Yes

Reviewer #2: (No Response)

4. Have the authors made all data underlying the findings in their manuscript fully available?

Reviewer #1: Yes

Reviewer #2: (No Response)

5. Is the manuscript presented in an intelligible fashion and written in standard English?

Reviewer #1: Yes

Reviewer #2: (No Response)

6. Review Comments to the Author

Reviewer #1: The authors responded appropriately to each of the observations made and provided well-supported justifications for their responses. I commend the authors for their work.

Reviewer #2: (No Response)

7. PLOS authors have the option to publish the peer review history of their article (what does this mean? ). If published, this will include your full peer review and any attached files.

**Do you want your identity to be public for this peer review?** For information about this choice, including consent withdrawal, please see our Privacy Policy .

Reviewer #1: **Yes: ** Irma Yvonne Amaya-Larios

Reviewer #2: No

---

## [Editor Report · Acceptance letter]

PONE-D-24-35337R2

PLOS ONE

Dear Dr. Bradfute,

I'm pleased to inform you that your manuscript has been deemed suitable for publication in PLOS ONE. Congratulations! Your manuscript is now being handed over to our production team.

Kind regards,

on behalf of

Dr. José Ramos-Castañeda

Academic Editor

PLOS ONE